# Human Pesticide Exposure in Bolivia: A Scoping Review of Current Knowledge, Future Challenges and Research Needs

**DOI:** 10.3390/ijerph21030305

**Published:** 2024-03-05

**Authors:** Jessika Barrón Cuenca, Kristian Dreij, Noemi Tirado

**Affiliations:** 1Genetic Institute, Medicine Faculty, Universidad Mayor de San Andrés, Av. Saavedra 2246 Miraflores, La Paz, Bolivia; jxbarron@umsa.bo; 2Institute of Environmental Medicine, Karolinska Institutet, SE-171 77 Stockholm, Sweden; kristian.dreij@ki.se

**Keywords:** pesticide exposure, health effects, Bolivia, farmers, Bolivian regulations

## Abstract

Numerous studies have shown that pesticide exposure is linked to adverse health outcomes. Nevertheless, in Bolivia, where there is an increasing use of pesticides, the literature is sparse. To address knowledge gaps and guide future research in Bolivia, we conducted a scoping review spanning 22 years (January 2000 to December 2022). Our search identified 39 peer-reviewed articles, 27 reports/documents on Bolivian regulations, and 12 other documents. Most studies focused on farmers and revealed high pesticide exposure levels, assessed through biomarkers of exposure, susceptibility, and effect. The literature explored a range of health effects due to pesticide exposure, spanning from acute to chronic conditions. Many studies highlighted the correlation between pesticide exposure and genotoxic damage, measured as DNA strand breaks and/or micronuclei formation. This was particularly observed in farmers without personal protection equipment (PPE), which increases the risk of developing chronic diseases, including cancer. Recent findings also showed the alarming use of banned or restricted pesticides in Bolivian crops. Despite existing Bolivian regulations, the uncontrolled use of pesticides persists, leading to harmful health effects on the population and increasing land and water pollution. This review underscores the need for the stringent enforcement of regulations and continued research efforts, and it provides a scientific foundation for decision-making by relevant authorities.

## 1. Introduction

Pesticides are the most used chemicals globally in agriculture, public health, and in domestic applications; consequently, a great part of the population may be exposed to these compounds.

Due to their potential toxicity, the use and handling of pesticides entails various risks for exposed workers, the general population, and the environment. The associated negative human health effects of pesticides can be both short and long term, and according to the World Health Organization (WHO), these chemicals are responsible for nearly one million accidental acute poisonings per year, of which 70% are occupational [1].

Since the late 1940s, when the “green revolution” was introduced in the Latin America and Caribbean (LAC) region, which was an industrialized agricultural exploitation model based on the application of synthetic agrochemicals, the utilization of agrochemicals, including pesticides, has increased dramatically in parallel with increased population and agricultural production [2]. Today, it is estimated that the LAC region accounts for 20% of worldwide consumption of pesticides and that more pesticides are used in Central and South America on a per capita basis compared to other regions in the world [3]. In addition, there is a lack of regulation (and its implementation) and education on safe pesticide use and handling to prevent severe adverse effects in pesticide users, consumers, and the environment, which together results in environmental pollution and risks to human health in this region.

In Bolivia, the use of agrochemicals began in the 1950s, promoted by North American food security aid programs, with donations of large quantities of chemical inputs, mainly organochlorines, substances that, due to their environmental and health effects, are prohibited today. It has been estimated that in the last 20 years, pesticide imports increased from 30 million kg in 2000 to 174 million in 2020 [4]. Most of the legally and illegally imported pesticides, which enter into Bolivia, come from China [5], representing a value of around USD 97 million in 2019. The Bolivian Ministry of Health confirms that “Pesticides, currently, are the dominant way to combat pests and diseases in the agricultural sector”. As in other LAC countries, the intensive and indiscriminate use of pesticides in Bolivia has likely resulted in widespread chronic human exposure [6].

Elevated exposure levels are of special concern for Bolivian agricultural workers who apply pesticides and are thereby exposed to different pesticides, e.g., organophosphates and organochlorines with established adverse health effects together with, e.g., herbicides like glyphosate, for which long-term health effects is not as well established [7,8]. As in many other low-to-middle-income countries, Bolivian farmers may mix many different pesticides when spraying their crops, and, most of the time, do so without the necessary PPE [8,9,10,11]. Moreover, Bolivian farmers who receive training on the use of PPE, proper pesticide handling, and integrated pesticide management (IPM) are less frequently poisoned by pesticides and report fewer symptoms after pesticide handling [12,13].

A recent scoping review of the health effects of pesticide exposure in LAC populations concluded that there is evidence that exposure to pesticides may cause health effects in these populations [3]. However, the authors also emphasized that most of the evidence was based on research performed in Brazil and Mexico, and that efforts should be made to support more widespread investments in research capacities in this region to better characterize the situation in other countries of the LAC region. A similar conclusion was made in a recent critical review on the occurrence and levels of legacy and emerging pollutants in the environment of the LAC region [14]. To address existing gaps of knowledge and to guide future research needs in Bolivia, we conducted a scoping review of the current available research on the health effects of pesticide exposure in Bolivian populations, as well as regulations.

## 2. Methodology

The literature search was conducted in PubMed, Web of Science, Scielo, and Google Scholar using the following keywords: “pesticide”, “herbicide”, “insecticide”, “fungicide”, “Bolivia” using the Boolean operator “AND”, and for studies published between January 2000 and December 2022. To identify literature in Spanish, the keywords “plaguicidas”, “herbicidas”, “insecticidas”, “fungicidas”, “agroquímicos” were used for the sources above in the same manner. Studies were included if they presented data related to human use, handling, and exposure to pesticides, as well as resulting human health effects. Additional relevant literature was identified by looking at the references of the initially identified literature. In total, 39 peer-reviewed articles were included and served as information sources. In addition, 27 reports and documents related to Bolivian regulations and international conventions, as well as 12 press releases and theses concerning agrochemicals, were identified via targeted manual searches of the websites of Bolivian and international institutions.

## 3. Pesticides in Bolivia

### 3.1. International and Local Regulations and Agreements

In order to guarantee its safe and responsible use of pesticides, Bolivia follows various international standards and agreements, such as the Rotterdam and Stockholm conventions [15,16,17], as well as guidelines and principles established by the Food and Agriculture Organization of the United Nations (FAO) for the safe and effective handling of pesticides [18,19], the World Health Organization (WHO) list of highly hazardous pesticides [20], and the Codex Alimentarius Commission related to food safety and quality, including levels of pesticides in food [21]. Furthermore, short-term pesticide intake ought not to exceed acute reference doses, and long-term pesticide intake should be below the acceptable daily intake limits recommended by the FAO/WHO Joint Meetings on Pesticide Residues to avoid negative health effects [22]. Moreover, article 16 of the Bolivian Political Constitution says that the State must guarantee diet security through healthy, adequate, and sufficient foods for the entire population.

In 2005, a government resolution marked a significant step forward with the publication of a list of prohibited pesticides to safeguard against the deleterious effects these substances can have. The list comprised ten pesticides, including several organochlorine pesticides (OCPs), selected based on their potential to pose risks and cause harm to human health due to their persistence and propensity for bioaccumulation within the environment, i.e., Persistent Organic Pollutants (POPs). Most of them were banned in 2002, but DDT was already banned in 1996. Later on, in 2015, endosulfan, monocrotophos, and methamidophos were banned, and the use of methomyl was restricted (Table 1) [23]. Surprisingly, these last two pesticides were recently reported as the most commonly used in some studies performed in Bolivian populations by Jors et al. [24]. and Barrón et al. [10].

With the objective of monitoring and further regulating the import of pesticides, the National Pesticide Technical Committee for the Registration and Control of Chemical Pesticides for Agriculture was established in 2016 [25]. To further improve the regulation, the National Service for Agricultural Health and Food Safety (SENASAG) approved an administrative resolution in 2018 pertaining to the “Regulations for the Registration and Control of Pesticides for Agricultural Use” that was based on provisions outlined in Decision 804 of the Andean Community of Nations and the Andean Technical Manual for the Registration and Control of Chemical Pesticides for Agricultural Use [26]. This new regulation introduced a three-tier evaluation process by various state entities for the import of pesticides into the country. This process encompasses several criteria, including an agronomic assessment conducted by SENASAG, a toxicological evaluation performed by the Ministry of Health, and an ecotoxicological evaluation carried out by the Ministry of Environment and Water.

In April 2022, through Supreme Decree No. 4702, the tariff tax was temporarily suspended, setting it at a zero percent (0%) rate for a span of around 7 months, up until 31 December 2022 [27]. This measure was introduced with the explicit goal of facilitating accessibility to agrochemicals for farmers and producers who were affected by the outbreak of the COVID-19 pandemic. By making agrochemicals more affordable, this initiative aimed to fortify food security while upholding national sovereignty and, additionally, to stabilize prices of essential goods within the family food basket.

### 3.2. Non-Governmental Organizations (NGOs) Engaged in Pesticide Management

In Bolivia, there are several NGOs that are responsible for promoting and executing programs or projects that provide advice on the proper handling and use of pesticides, as well as agroecology, and, in this way, they reduce the existence of possible risks of damage to human health and the environment.

Plagbol: a Bolivian NGO whose purpose is to work with the problem of the use and management of pesticides and other chemical contaminants. From 2005 to 2010, Plagbol performed research projects with the support of the Danish development cooperation (Danida) together with researchers at Universidad Mayor de San Andrés in La Paz that were related to occupational exposure to pesticides, looking for health effects due to exposure [24,28,29,30]. This NGO has provided training and advice on IPM and the adequate use of personal protection equipment for farmers. Nowadays, Plagbol also provides training in production and the marketing of quality fruit and vegetable products, as well as evaluations of social and productive projects [31].

GTCCJ: a Working Group on “Climate Change and Justice”. In one of its thematic axes, this NGO works together with other institutions in actions that contribute towards sustainable food alternatives, such as agroecological production, food education, solidarity economy, and responsible consumption. GTCCJ developed a study on the Use and Management of Agrochemicals in Agricultural Production, under the auspices of MISEREOR and Pan for the World, managed by the NGO INCADE, together with the Research Institute of the Faculty of Humanities and the Gabriel René Moreno Autonomous University in Santa Cruz de la Sierra. The results revealed an excessive use of agrochemicals, including red labeled ones (highly toxic pesticides), and others that are prohibited in the national and world markets [32].

Probioma: Productivity Biosphere Environment is a private social development institution based in Santa Cruz. Founded in 1990, it has extensive experience in the areas of agroecology, biodiversity management, biotechnology, strengthening the capacities of local organizations, training of socio-environmental monitors, political advocacy, and citizen information. Probioma promoted and supported the bill to promote the manufacture and use of agroecological bio inputs in food production and environmental bioremediation, establishing the guidelines of the National Agricultural bio inputs policy to guarantee the sovereignty and food security of the country [33].

Agrecol Andes: This Foundation has been working for more than 20 years, promoting and executing a proposal for sustainable productive development, which is framed in various agroecological plans, programs, and projects. Agrecol’s work area is focused on rural, urban, and peri-urban areas of the department of Cochabamba, but through networks and platforms, it promotes actions in other departments of the country, including the Andean-Amazon region, together with other partners and target groups. Some relevant work strategies of Agrecol are as follows: Contribute to a healthy and responsible diet, with organic products accredited by the Participatory Guarantee System (SPG), which is covered by the National Technical Standard of Law 3525, and other forms of ecologically guaranteed alternatives. Develop research processes, systematization, and dissemination of the ancestral knowledge of the original indigenous peoples, on issues of agroecological development, the comprehensive management of water resources, the management and care of water, healthy eating, short marketing circuits, and the generating of knowledge to share with different people. Develop actors such as community organizations; rural, urban, and peri-urban families; authorities of autonomous territorial entities; and private sector development organizations [34].

Solon Foundation: works with the research and production of audiovisual, printed, and digital materials on different subject areas such as nature, energy, law and regulation, and economy, among others. In the area of nature, the topics of forests are included, as well as livestock, transgenics, agrochemicals, water, and climate change. The materials referring to the topic of agrochemicals include reviews of import regulations for pesticides, banned pesticides, agroindustry, and the health effects of pesticides [35,36,37,38].

### 3.3. Current Use and Import of Pesticides

Agriculture plays a pivotal role in Bolivia, constituting the second most significant source of income after mining. The Bolivian Andes region, situated at elevations exceeding 2000 m above sea level with temperatures ranging from 17 °C to 27 °C, provides favorable conditions for cultivating a variety of vegetables such as lettuce, tomatoes, pumpkins, and broccoli, as well as fruits like pears, apples, figs, and coca leaves, among others. Conversely, the eastern part of the country, situated at elevations between 500 to 700 m above sea level, experiences higher temperatures ranging from 28 °C to 35 °C or more, which allows for the cultivation of crops such as citrus fruits, watermelons, pineapples, soybeans, corn, and sugar cane, among others. The broad range in elevations and temperatures allows for year-round harvests of diverse crops, albeit contributing to an increased reliance on pesticides for crop protection [39].

Since Bolivia does not produce but only imports pesticides, not only large-scale but also small-scale farmers have become target clients of massive marketing campaigns from pesticide importing companies. Indeed, according to FAO statistics, Bolivia’s pesticide usage per unit of cultivated area doubled from 1.86 kg/ha in 1997 to 3.29 kg/ha in 2017 [40]. Other data of total pesticide use measured in tons shows that Bolivia used, in 2020, 19,295 t, more than Perú with 10,631 t and Chile with 9831 t but less than Argentina with 241,294 t and Brazil with 377,176 [41]. Bolivia depends entirely on pesticides imported from other countries, mainly from China, Uruguay, Paraguay, Brazil, Argentina, and Peru [37]. This importation is carried out both legally and illegally. It is estimated that between 14 and 35% of pesticides sold nationally are smuggled [5]. Thus, this large number of illegal pesticides means that there is significant under-recording; therefore, official import and use data represent just a fragment of the status of pesticides in the country [6].

Recent data on pesticide import and use are available from SENASAG. In 2019, the government permitted the import of more than 7000 tons of various agrochemical products, glyphosate, and its various formulations, being the most imported pesticides [36]. In 2020, 82 companies were registered to be engaged in the import of pesticides and related chemical substances, with most of the agrochemicals imported from China [5,42]. The same year, SENASAG registered 2120 agricultural input products throughout the country, of which 1863 were chemical pesticides for agricultural use. These pesticides mainly represented three categories: herbicides (35.5%), insecticides (33.3%), and fungicides (27.8%) [43].

According to the Administrative Resolution (RA) No. 055/2002 of SENASAG, which regulates the toxicological classification of pesticides based on the Recommended Classification of Pesticides by hazard in accordance with the WHO (2020) [44], there are five classification categories: Extremely hazardous (Ia), Highly hazardous (Ib), Moderately hazardous (II), Slightly hazardous (III), and those that normally do not offer danger are classified as Unlikely to present acute hazard (U). Of these, pesticides categorized as Extremely hazardous (Ia) are restricted, but not prohibited, by RA No. 055/2002 Article 30 and may still be subject to risk–benefit studies by the registrant company [8]. Of the 1863 chemical pesticides registered by SENASAG in 2020, 43.4% corresponded to category II, 30.9% to category IV, and 23.7% to category III. The remaining 2% (37 records) belonged to category Ib pesticides (Figure 1). The five active ingredients with the highest number of registrations until the end of 2020 were glyphosate, azoxystrobin, thiamethoxam, imidacloprid, and paraquat, all belonging to category II. The large use of category II pesticides was also observed in a survey on the use of pesticides in Bolivia in 2018 and 2020 [10,11]. A detailed review of the SENASAG database indicated the presence of about 200 active ingredients. Of all these, the most recurrent was glyphosate, which had 121 different registered products. In second place was azoxystrobin with 77 records, followed by thiamethoxam, imidacloprid, and paraquat with 71, 65, and 54 registered products, respectively (Table 2).

## 4. Human Exposure in Bolivia

Few studies have measured exposure levels in Bolivian populations, which to some degree, probably relates to the advanced methodology required [3]. Human biomonitoring of pesticide exposure is done in, e.g., urine or blood, where either the parent chemical or its metabolites are measured by liquid or gas chromatography [45]. Similarly, few environmental monitoring studies performed in Bolivia were found, which can be performed in, e.g., air, water, and food to assess levels in possible sources of exposure.

### 4.1. The general Population

Although the use of organochlorine pesticides (OCPs, e.g., DDT) has been banned in Bolivia for many years, due to their persistence, they can bioaccumulate in humans and be a potential health concern. An analysis of 112 breast milk samples from pregnant women visiting a hospital in the city of El Alto showed that 62 samples contained detectable levels of dieldrin, HCB, lindane, and DDT (measured as p,p′-DDT, o,p′-DDE, and p,p′-DDE), and 13 samples exceeded the maximum permitted total concentration of OCPs of 0.2 µg/g, while 5 samples exceeded the same limit value for total DDT [46]. Similarly, DDT (measured as o,p′-DDT, and p,p′-DDE) was detected in >80% of cord serum samples in a Bolivian birth cohort in the city of Santa Cruz (*n* = 200), with mean concentrations of 39.5 and 196.7 ng/g lipid [47]. The presence of DDT (measured as p,p′-DDT and p,p′-DDE) and HCB was also found in serum and adipose tissue collected from an adult cohort from the city of Santa Cruz [48,49]. p,p′-DDE showed the highest geometric mean adipose tissue concentration (386.6 ng/g lipid) and highest serum concentration (267.4 ng/g lipid). In comparison, HCB mean concentrations in adipose tissue and serum were 26.3 and 22.1 ng/g lipid, respectively. These concentrations were similar to those reported from countries that had completely banned the use of OCPs. Combining questionnaire data and concentration data by multiple linear regression models, age, diet, and smoking habit were identified as the main exposure predictors of p,p′-DDE.

In terms of environmental monitoring, measurements of air concentrations of new and legacy POPs, including OCPs, have been performed in Bolivia within the Global Atmospheric Passive Sampling network [50,51]. Measurements performed on the east side of the Andean Mountain range of Bolivia for one year consistently detected insecticide hexachlorocyclohexanes and endosulfans, while, for example, p,p′-DDT was only detected twice. The highest concentrations were found for endosulfans (up to 1751 pg/m^3^) between February–June, which coincides with the period of high agricultural activity. Based on the chemical analysis, the authors concluded that both local and regional emissions contribute to air levels of OCPs and that such air pollution could be a likely source of exposure for populations in Bolivia. The general trend for the entire LAC region is, however, decreasing levels in the air of these two groups of pesticides [51].

Levels of pesticides have also been investigated in food. A study carried out in Omereque and Rio Chico detected six banned organochlorine (e.g., aldrin and heptachlor) and five organophosphate (e.g., chlorpyrifos and methyl parathion) pesticides in tomatoes [52]. Of the OCPs, only heptachlor was consistently detected, up to 63 µg/kg. Organophosphates were found at much higher levels, with, for example, methyl parathion and malathion at above 1 mg/kg. Washing and peeling tomatoes was also shown to reduce pesticide levels very efficiently (50–100%). The results from this case study revealed a risk associated with the consumption of Bolivian tomatoes, particularly for children, and mostly associated with the organophosphate residues while the risk linked to OCPs was minor. Of most concern was the amounts of methyl parathion that were above the maximum permitted limit (0.2 mg/kg) [53]. This pesticide is extremely hazardous according to the WHO (class Ia) and may cause sweating and involuntary contractions when in contact with the skin, as well as affect the central nervous system in cases of acute poisoning. A follow-up study analyzed almost 300 pesticides in 10 potatoes, onions, and lettuce obtained from different markets in the city of La Paz. No pesticides were found in the two former vegetables and only cypermethrin, chlorpyrifos, difenoconazole, and λ-cyhalothrin were detected in 5 lettuce samples, up to 2 mg/kg. Only the levels of chlorpyrifos exceeded the maximum allowed limit, and the authors concluded that these levels were not associated with increased acute or chronic health effects [54].

In summary, the few existing studies showed a potentially high exposure level for the general Bolivian population to OCPs due to historical use. Of extra concern are the data suggesting that children could be exposed to elevated levels of organophosphates via vegetables. These studies should be repeated and expanded to assess the current exposure levels in the general population, especially for susceptible groups such as pregnant mothers and children.

### 4.2. Agricultural Workers

In efforts to characterize human exposure to pesticides in Bolivia from different perspectives and extending the studies described above [47,48,49], serum concentrations of p,p’-DDE were analyzed in 70 agricultural workers from three rural communities (Algodonal, Aguas Claras, and La Junta) in Santa Cruz [55]. The results showed detectable levels of p,p′-DDE in all samples with median concentrations of 19.7 ng/mL (4789 ng/g lipid), which was comparable to levels found in populations with an ongoing exposure to DDT. As expected, the serum levels were substantially higher (around 20-fold) than those found previously by the same group in urban Santa Cruz populations. Since no data indicated the current use of DDT in these communities, the authors concluded that the high serum concentrations were likely due to a contaminated environment due to historical use in this region. This was further supported by a strong association between serum levels and time of residence in the study area.

Since several questionnaire-based studies had indicated the poor handling of pesticides as well as poor use of PPE among Bolivian agricultural workers, Barrón Cuenca et al. performed a cross-sectional study in three rural communities to better assess the impact of these behaviors on the level of pesticide exposure [10]. The study included 275 active farmers in Sapahaqui, Department of La Paz, Villa Bolivar, and Villa 14 de Septiembre in Chapare, Cochabamba. The handling of PPE, use of PPE, and pesticide exposure were assessed using a questionnaire and measurements of 10 urinary pesticide metabolites (UPMs). Methamidophos, paraquat, and glyphosate were the most frequently used pesticides, and most of the farmers combined several pesticides while spraying their crops, which was in agreement with previous Bolivian surveys [8,56]. The low number of farmers who used recommended PPE (17%) was also in accordance with what has been reported previously among Bolivian agricultural workers [56,57,58]. The highest mean urinary concentrations were detected for 3,5,6-trichloro-2-pyridinol, a metabolite of chlorpyrifos (17.6 ng/mL), and 2,4-dichlorophenoxyacetic acid (2,4-D, 15.8 ng/mL), although maximum concentrations could reach 100-fold higher. In general, men had higher urinary concentrations of pyrethroids and 2,4-D compared to women (*p* < 0.05). Regression analysis revealed that farmers who were better at following recommendations for pesticide handling and the use of PPE had a significantly lower risk of having high UPM levels of pyrethroids.

By measuring biomarkers of exposure, the studies by Mercado et al. [55] and Barrón Cuenca et al. [10] were the first studies showing that agricultural workers in Bolivia are exposed to high levels of pesticides. Moreover, the latter study extended and confirmed previous qualitative Bolivian studies and emphasized the importance of training in proper protection and pesticide handling to reduce exposure levels in these populations. Such activities should be further promoted through activities by national authorities and NGOs to protect farmers’ health and secure safe food production.

### 4.3. Health Workers in Vector Control Programs

In Bolivia, Chagas disease remains a major public health risk [59]. Chagas is caused by the parasite *Trypanosoma cruzi* and is transmitted by triatomine bugs and may cause congestive heart failure if left untreated. To combat this disease, indoor residual spraying, which, since the 1980s, mainly involves pyrethroids (e.g., cypermethrin), is conducted by health workers in a public vector control program [60,61]. Such programs are conducted globally to minimize the spreading of several tropical diseases, including malaria and dengue. Spraying workers may spend long hours indoors applying large quantities of pesticides using handheld equipment. It is thus of high importance to follow national and international guides and regulations for proper pesticide handling and use of PPE to reduce the level of exposure [19,62]. Hansen et al. assessed the exposure and spraying practices of 120 workers of the Bolivian vector control program using questionnaire data [28,29]. The median number of years employed as a pesticide sprayer was 10 years, and the number of hours spraying per week ranged from 21 to 56 h. About half of the workers had only sprayed with pyrethroids in their occupation. Notably, the results showed that 26% had not received any training in pesticide handling, and only 18% had taken a course in the last year. The low level of training was reflected in their spraying practices; >85% of the sprayers never wore boots with high shafts, aprons, or rubber gloves, and 43% sometime ate or drank during spraying. Around a third of the workers only sometimes washed themselves and changed clothes after spraying. Although the situation of the vector control sprayers in Bolivia has not been extensively studied, these results indicate that not only agricultural workers but also other occupations that use pesticides would benefit from improved training.

## 5. Health Effects in Bolivian Populations

Based on the studies above, it is clear that some Bolivian populations handle and may be exposed to high levels of pesticides. Our literature search identified a number of studies that studied health effects due to pesticide exposure, encompassing a spectrum of issues ranging from acute to chronic health effects.

### 5.1. Acute Pesticide Poisoning (APP) Due to Suicide Attempts

In a one-year descriptive study conducted at the Emergency Department of the Hospital de Clínicas in La Paz, a total of 300 cases involving intentional ingestion of organophosphate or carbamate pesticides were examined. Among these cases, 97% were identified as suicide attempts or suicides. This phenomenon was more prevalent in females than males, with a ratio of 2:1. The most common health symptom reported when they arrived at the emergency unit was abdominal pain (83%), followed by nausea/vomiting (79%), and miosis (72%), among other symptoms. The study concludes that although the mortality rate was only 6%, complications such as aspiration, cardiopulmonary arrest, and seizures were the most frequent outcomes [63].

Similar findings emerged from a five-year study that conducted a comparative analysis of suicide attempts and completed suicides concerning age and gender in relation to pesticide ingestion. The study drew upon four distinct sources of information: emergency room and psychiatric records, national crime statistics, and newspaper articles, yielding a comprehensive dataset comprising 1124 cases. It was found that suicide attempts displayed a higher incidence among young women (61%) as compared to men (39%) (*p* < 0.05). However, for completed suicides, a larger proportion of men (70.5%) were involved. Notably, the data underscored the finding that adolescents constituted the age group with the highest frequency of suicide attempts, but that completed suicides were more prevalent within the older age group [64].

### 5.2. Occupational Acute Pesticide Poisonings

A study conducted across three agricultural communities in Bolivia revealed a direct correlation between the improper use of PPE during pesticide spraying and the heightened risk of exposure and associated APP incidents. An evident gender disparity emerged in PPE utilization, with significantly fewer women conforming to adequate protective measures compared to their male counterparts (*p* < 0.01). This gender-specific discrepancy was mirrored in the prevalence of APP symptoms during pesticide application. The findings revealed that 80% of farmers experienced signs or symptoms of APP, with women notably more affected than men (*p* < 0.05). The most commonly reported symptoms among women included headache, nausea, vomiting, and fasciculations. Men predominantly experienced symptoms such as dizziness (*p* < 0.05) and red eyes [10]. Moreover, about 44% of the study population reported symptoms associated with pesticide use in a cross-sectional study conducted within a total of 50 families from Punata, Cochabamba [65]. Both studies emphasize the urgency of addressing gender-related discrepancies in PPE compliance and APP occurrences within these communities.

A separate investigation highlighted the evident influence of gender disparities, knowledge levels, and pesticide handling practices on the prevalence of APP symptoms among women farmers from La Paz. These outcomes were accentuated by lower educational backgrounds [57]. This phenomenon was further underlined by a parallel study conducted by the same group of researchers. This study showed that the absence of PPE, inadequate knowledge, and unsatisfactory hygienic and hazardous practices while dealing with pesticides exerted a substantial impact on the heightened likelihood of APP symptom manifestation. However, these factors did not correlate with serum cholinesterase levels during pesticide handling [56].

In summary, women are clearly a vulnerable group in relation to APP, either as a result of intended or unintended occupational exposures. More efforts need to be taken to promote mental health as well as improve training in the proper handling of pesticides for women to reduce the high incidence of APPs.

### 5.3. Genotoxic Effects

Numerous studies have been published concerning the correlation between pesticide exposure and the potential for genotoxic damage across different Bolivian populations. Most studies report either levels of DNA strand breaks (by the Comet assay) and/or formation of micronuclei (by the Micronuclei assay) in peripheral blood lymphocytes and/or oral mucosa.

As part of PLAGBOL, an investigation was conducted to assess the genotoxic effects of pesticides among a cohort of 81 volunteers from La Paz. This group encompassed 48 farmers with direct pesticide exposure and 33 non-exposed individuals serving as controls. The findings showed a significative increase in genetic damage among pesticide-exposed farmers when compared to the control group (*p* < 0.001) and that living altitude was an independent risk factor for DNA damage in lymphocytes. Additionally, a positive correlation between the intensity of pesticide exposure and the frequency of chromosomal aberrations was observed, particularly in the male participants [24].

In a Bolivian thesis, genotoxic damage was measured in 118 occupationally exposed farmers and 80 non-farmers. Significant increased levels of genotoxic damage (*p* < 0.05) were observed in farmers exposed to pesticides, particularly those working with only organophosphates and those who combined pyrethroids and organophosphates, compared to the control population. The increased damage was most prominent among farmers who lacked PPE, had insufficient knowledge regarding pesticide handling, and accumulated more years of pesticide use [7]. In agreement with this, another study showed that farmers exposed to pesticides who lacked proper protection and safety measures were 1.49 times more likely to experience genotoxic damage, yielding an odds ratio (OR) of 2.49 (CI: 1.48–4.20) [66]. Another research team observed genotoxic damage among children under three years of age afflicted with chronic malnutrition. These children encountered pesticide exposure through direct contact with their mothers. Despite statistical insignificance, the researchers inferred that these children exhibited a heightened vulnerability to develop chronic and degenerative illnesses in the future [67].

In a study performed in 297 participants from three agricultural Bolivian communities, levels of genotoxic damage were subject to various influencing factors, including years of experience and the use of PPE. The outcomes indicated a greater frequency of micronuclei (MN) in both women and men engaged in farm activities for a time more or equal to 8 years, compared to their respective counterparts (*p* < 0.05). Moreover, those researchers conducted a measurement of ten UPMs and employed logistic regression techniques. This analysis revealed a notable increase in the risk of DNA strand breaks for individuals with substantial exposure to 2,4-D, yielding an OR of 1.99 (*p* < 0.05). Similarly, an elevated incidence of genotoxic damage in individuals with high exposure to tebuconazole, 2,4-D, or cyfluthrin (*p* < 0.05–0.001) was found. As a reflection of the common practice among Bolivian farmers to combine multiple pesticides in a single application, the same study observed that heightened exposure to specific pesticide mixtures, primarily containing 2,4-D or cyfluthrin, correlated significantly with heightened levels of genotoxic damage and associated increased risk (OR = 1.99–2.94, *p* < 0.05). To investigate the impact of PPE use and pesticide handling on the risk of having increased UPM concentrations, a “protection and handling index” score (PHI score) was created for each participant, based on the guidelines provided by the FAO. Even though a protective effect against exposure was found in farmers who were good at following recommendations, the PHI score did not show an association with reduced levels of genotoxic damage [68].

Furthermore, a number of studies have investigated the impact of polymorphisms in cytochrome P450 (CYP) and glutathione transferase (GST) genes on genotoxicity among Bolivian farmers exposed to pesticides. These enzymes play vital roles in both activating and detoxifying substances with the potential to act as either risk factors or protective agents in cancer development. The impact of polymorphisms in the CYP1A1 gene on DNA damage was explored in 92 farmers exposed to pesticides in a Bolivian thesis. The study found that CYP1A1 genotypes labeled as risk factors (m1wt) are not directly associated with cancer. However, a significant association emerged in individuals who smoked or consumed alcohol, indicating higher risk values than those attributed to individual factors. This underscores the importance of considering environmental factors in risk assessment. Notably, despite alcohol not being a substrate for CYP1A1, a connection was found, possibly due to overlapping habits among smokers and alcohol consumers, creating a complicating autocorrelation effect. Additionally, the authors noted that alcohol enhances the formation of benzo[*a*]pyrene adducts, offering a partial explanation for the observed association [69].

In a study involving 142 individuals from two small agricultural communities in La Paz (99 exposed farmers and 43 controls), the impact of GST M1 and T1 polymorphisms on cholinesterase activity, sister chromatid exchange, and micronuclei frequency was explored. The GSTM1 null genotype showed lower micronuclei frequency compared to GSTM1 positive, although it was not statistically significant (*p* = 0.075). The GSTT1 null genotype exhibited a micronuclei frequency similar to the GSTT1 positive genotype. When evaluating the association between GST polymorphisms and genotoxic damage, it was found that the GSTM1 null population carried a 0.38 times higher risk of experiencing genotoxic damage compared to individuals with the gene [70]. Similar to this, in the Bárron Cuenca et al. study [68], the majority (54%) exhibited the GSTM1 null genotype, while a significant portion (69%) showed the GSTT1 positive genotype. Individuals with the GSTM1 null genotype had higher DNA strand break levels compared to positive genotypes (*p* < 0.05 for tail moment; *p* = 0.067 for %DNA in tail). Similarly, for GSTT1, DNA strand break levels were higher in the null group, though not statistically significant. Those null for both GST genotypes displayed elevated DNA strand break levels, particularly in tail moment (*p* < 0.05). Notably, individuals with positive GSTM1 showed a higher MN frequency (*p* < 0.05), aligning with the previous study [70].

In summary, all the identified studies demonstrate a significant correlation between handling or pesticide exposure and increased levels of genotoxic damage among Bolivian farmers, especially among those with high exposure or many years or farming. What is lacking are longitudinal studies where exposure and genotoxicity are followed over time to make stronger associations as well as provide better information about cancer incidences in the more rural agricultural regions of Bolivia.

### 5.4. Other Health Effects

One of the main health concerns of OCP exposure is developmental effects such as fetal growth impairment due to their endocrine-disrupting properties. The association between prenatal DDT exposure and birth outcomes was investigated in a birth cohort from Santa Cruz [47]. The study included 200 mothers and their newborns. DDT exposure was assessed from the cord blood serum levels of o,p′-DDT and p,p′-DDE and birth outcomes by anthropometric measurements and gestational length. Based on multivariable regression analyses, opposite associations with birth weight were found for p,p′-DDE (β = 0.012, *p* = 0.006) and o,p′-DDT (β = −0.014, *p* = 0.039). In addition, the higher cord blood of p,p′-DDE was negatively associated with gestation time (β = −0.004, *p* = 0.012). These mixed associations are in agreement with those reported from other LAC countries [3].

Pyrethroids, which are among the most used insecticides both in Bolivia and worldwide, are less toxic to humans compared to organochlorine insecticides but, regardless, are associated with neurotoxicity and reproductive toxicity [71]. In the Bolivian population, potential chronic health effects of pyrethroids have only been studied in spraying workers in the public health vector control program, which mainly uses pyrethroids (see Section 4 above). To assess the risk of developing diabetes in this occupational group, the association between cumulative pyrethroid exposures and levels of glycosylated hemoglobin was studied in 116 pesticide sprayers and 92 non-exposed controls [29]. Duration, intensity, and cumulative exposures were assessed from questionnaire data. The results showed no clear dose-dependent correlation between pyrethroid exposure and abnormal glucose regulation. However, a statistically significant trend was observed between cumulative exposure and OR of abnormal glucose regulation for sprayers who had only used pyrethroids (*p* = 0.01). The authors cautioned the interpretation of these results since the control group was quite different from the sprayers (e.g., age, BMI, tobacco use).

The same sprayer population was also assessed for the occurrence of neurological deficits [28]. The endpoints included neuromotor and neurocognitive performance and subjective CNS symptoms (blurred vision, headache, etc.). A significant increased prevalence of subjective CNS symptoms was reported by workers with higher spraying intensity and higher cumulative exposure compared to lower quintiles of exposure (OR = 1.92 and 2.01, respectively). No associations were observed with effects on neuromotor performance, but higher spraying intensity was correlated with significantly reduced neurocognitive performance (adjusted β per quintile = −0.405). The authors concluded that the pyrethroid exposure of these workers may cause chronic health effects and that education and training of sprayers in the public health vector control program in the proper handling of pesticides is important to reduce these health risks. This is in accordance with a recent systematic review, which concluded that a majority of the studies included showed an increased risk of neurological effects from pyrethroid exposure in agricultural workers and their children [72].

## 6. Future Challenges and Research Needs in Bolivia

The findings from the reviewed studies provide strong evidence supporting the assertion that pesticide exposure leads to adverse health effects, including genotoxic damage and degenerative diseases in the Bolivian population. Addressing this issue requires a focus on biomonitoring, a crucial foundation for effective medical surveillance. This approach aids in evaluating potential risks from both occupational and accidental exposures, facilitating the early identification of health risks.

To reduce the illegal importation of pesticides, it is imperative to strengthen legislation and enhance border controls, monitoring local fairs and retail stores. This proactive strategy aims to prevent the use of highly toxic, restricted, and unauthorized pesticides. Additionally, a comprehensive review of legislation is necessary, aligning it with current recommendations from reputable organizations such as the WHO, the Pesticide Action Network, Greenpeace, and the International Code of Conduct for Pesticide Management by the WHO and FAO.

Prioritizing organic agriculture in Bolivia requires concerted efforts in training, research, and agricultural guidance. Supporting organic ventures within diversified production systems and ensuring compliance with Law 3525, which outlines responsibilities for promoting and controlling organic production, is crucial.

Raising awareness and adopting responsibility among corporations and individual farmers in pesticide management and proper waste disposal are essential. Farmers adhering to these principles could obtain certifications validating their adherence to established standards for healthy agricultural production.

Introducing agricultural practices that minimize pesticide use, such as crop rotation, cover crops, polycultures, and integrated pest management, is essential. Encouraging the use of protective clothing and equipment, supported by educational campaigns, would serve as a potent tool to raise awareness about the acute and chronic health risks associated with pesticide exposure, particularly with highly toxic pesticides.

The insights drawn from the studies reviewed here could serve as a scientific basis for authorities to fund rigorous ecological and epidemiological studies. Such an initiative would enhance surveillance systems’ understanding of the detrimental effects of these substances on health and the environment. Ultimately, these efforts aim to empower the government to make informed decisions that improve the quality of life and food security for Bolivians.

Beyond doubt, it is important to remark that we need more investigations about the consequences of the use and abuse of pesticides, not only in populations occupationally exposed but also the general population; women, residents in adjacent areas, and children must be included since they are indirectly exposed to pesticides by other sources such as water and contaminated food. We strongly want to emphasize that the government should support this kind of research to keep our population safe and healthy.

## 7. Conclusions

This scoping review provides evidence that exposure to pesticides may have a negative impact on the health of Bolivian populations. It establishes the scientific basis for authorities to make informed decisions, emphasizing the importance of respecting, safeguarding, and advancing human rights related to health, nutrition, and a clean environment. This is an essential ethical and legal commitment for all Bolivians.

More studies related to pesticide use and its consequences are still needed. Considering the health implications, it is important to introduce safer and ecological agricultural practices that reduce the use of pesticides. In addition, the use of protective measures for workers needs to be improved by joint efforts from authorities and NGOs in informing and training farmers and spray workers.

Finally, we conclude that prioritizing organic agriculture in training, research, and agricultural guidance is of the utmost significance. Multisectoral approaches, such as a closer collaboration between researchers, farmers, and policymakers to develop recommendations and interventions, can be effective in promoting food security, public health, and environmental conservation. The shift towards organic agriculture can also contribute to these goals by reducing reliance on harmful chemical inputs, not only in Bolivian agriculture but also worldwide.

## Figures and Tables

**Figure 1 ijerph-21-00305-f001:**
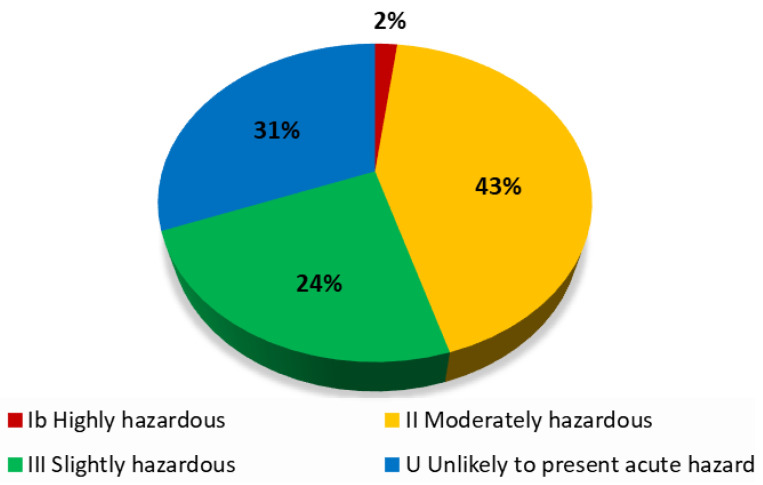
Grouping of pesticides registered in Bolivia in 2020 based on their WHO hazard classification.

**Table 1 ijerph-21-00305-t001:** Pesticides that are prohibited or restricted in Bolivia and their technical justifications, adapted from the “List of prohibited and restricted pesticides in Bolivia” published by the Bolivian Ministry of Environment and Water [23].

PROHIBITED PESTICIDES	TECHNICAL JUSTIFICATION
Dieldrin	Listed in the Stockholm Convention [16]. Prohibited since they are POPs; their use can cause risks and damage to human health and the environment because of their high persistence and bioaccumulation properties.
Endrin	
Toxaphene	
Mirex	
Dichlorodiphenyltrichloroethane (DDT)	
Chlordane	
Hexachlorobenzene (HCB)	
Aldrin	
Heptachlor	
2,4,5-Trichlorophenoxyacetic acid	
Endosulfan	
Monocrotophos Methamidophos	For being highly toxic to human health and for causing damage to the environment. Included in Annex III of the Rotterdam Convention [15].
**RESTRICTED PESTICIDES**	**TECHNICAL JUSTIFICATION**
Methomyl	Harmful to health and the environment. Its import, commercialization, and use are authorized under a prescription prescribed by an agronomist accredited by SENASAG.

**Table 2 ijerph-21-00305-t002:** List of active ingredients in pesticides registered by the SENASAG in 2020.

N°	Active Ingredient	N° of Registered Products	Percentage	Activity	Used for	Toxicity Range Classification	N° of Countries Where Banned
1	Glyphosate	121	6.5%	Herbicide	Soy, fallow	II, III, IV	3
2	Azoxystrobin	77	4.1%	Fungicide	Other oleaginous, rice	II, III, IV	-
3	Thiamethoxam	71	3.8%	Insecticide	Soy, sorghum, corn	II, III, IV	28
4	Imidacloprid	65	3.5%	Insecticide	Soy, sorghum, corn	II, III, IV	28
5	Paraquat	54	2.9%	Herbicide	Soy, fallow	Ib, II, III	48
6	Emamectin benzoate	54	2.9%	Insecticide	Soy, corn, rice	II, III, IV	-
7	Mancozeb	50	2.7%	Fungicide	Tomato, potato, soy, wheat	III and IV	29
8	Abamectin	45	2.4%	Insecticide/ Acaricide	Soybean	Ib, II	-
9	Atrazine	44	2.4%	Herbicide	Corn, sugar cane	II, III, IV	41
10	2,4 D	42	2.3%	Herbicide	Fallow, soybean, rice, sugar cane	II; III	3
11	Tebuconazole	42	2.3%	Fungicide	Soybean, rice, corn	II, III, IV	1
12	Lambda Cyhalothrin	40	2.2%	Insecticide	Soybean, corn	Ib, II, III, IV	28
13	Carbendazim	35	1.9%	Fungicide	Soybean, corn	III, III, IV	32
14	Fipronil	35	1.9%	Fungicide	Soybean, beans, corn	II, III	37
15	Chlorpyrifos	33	1.9%	Insecticide	Soybean, corn, tomatoes	Ib, II	35
16	Clethodim	31	1.7%	Herbicide	Soybean, sunflower, fallow	II, III, IV	-
17	Lufenuron	26	1,4%	Insecticide	Soybean, corn, wheat	II, III, IV	28
18	Chlorantraniliprole	25	1.4%	Insecticide	Soybean, corn, rice	II, III, IV	-
19	Cyproconazole	22	1.2%	Fungicide	Soybean, corn, rice	II, III, IV	-
20	Bifenthrin	20	1.1%	Insecticide	Soybean, rice	Ib, III	29

Adapted from Villalobos 2021 (Radiografía de los agroquímicos en Bolivia), Solon Foundation [35].

## Data Availability

Not applicable.

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
