# Peer review of "Human Pesticide Exposure in Bolivia: A Scoping Review of Current Knowledge, Future Challenges and Research Needs"

_ijerph, 2024, doi:10.3390/ijerph21030305_

Round 1
Reviewer 1 Report
Comments and Suggestions for Authors
The authors present a review of review articles and other reports over a period of twenty years regarding the usage of persticides in Bolivia. They address a relevant and disconcerting issue.
The English language usage is fine and reads well.
The search criteria are lucid.
The manuscript reads more as a report than a scientific article. I find the discussion of the cited articles somwehat superficial and this probably explains why I find the manuscript more a report than a review article.
The first part of the results section reads more as a summary of the history of usage and control of pesticides in Bolivia. These aspects are probably better suited to a supplemental information section.
I would recommend more depth to the discussion of the cited literature.
Author Response
We are very grateful to you for the time invested in reviewing our manuscript and for the helpful comments and suggestions that helped us to clarify and generally improve our paper by pointing out missing information.
We have carefully considered each of your comments and modified the text to address the concerns raised. The guidance has undoubtedly enriched the overall content of the paper.
Please find our point-by-point responses below.
Reviewer 1:
The authors present a review of review articles and other reports over a period of twenty years regarding the usage of pesticides in Bolivia. They address a relevant and disconcerting issue.
Response 1: Thank you. We would however like to clarify that the review is based on original papers and reports and not a review of reviews.
The English language usage is fine and reads well.
Response 2: Thank you. Minor editing has been performed throughout the manuscript to further improve grammar and style.
The search criteria are lucid.
Response 3: Thank you for your comment.
The manuscript reads more as a report than a scientific article. I find the discussion of the cited articles somewhat superficial and this probably explains why I find the manuscript more a report than a review article.
Response 4: This is a valid point, but a scoping review is a relatively "new" approach to synthesizing the research literature that differs from the traditional critical or systematic review. The primary aim of a scoping review is to provide the reader with an overview of the current evidence from the literature on a particular research topic, without providing a summary answer or critical review to a discrete research question. Scoping reviews are as a result usually less comprehensive than systematic reviews. See e.g., Munn et al 2018, https://doi.org/10.1186/s12874-018-0611-x
We have, however, expanded on the discussion/summary of most of the different evidence found in our literature search, see pages 14, 16, 18, and 21.
The first part of the results section reads more as a summary of the history of usage and control of pesticides in Bolivia. These aspects are probably better suited to a supplemental information section.
Response 5: Thank you for your suggestion, but we think that it is important to give a background on pesticide use in Bolivia the history of usage and control of pesticides to understand the development into the current situation
I would recommend more depth to the discussion of the cited literature.
Response 6: See the above response 4.
Reviewer 2 Report
Comments and Suggestions for Authors
Dear Editor/Authors
The paper "Human Pesticide Exposure in Bolivia: A Scoping Review of Current Knowledge, Future Challenges, and Research Needs" provides a comprehensive overview of pesticide exposure in Bolivia and its various health impacts. The paper provides a comprehensive review of pesticide exposure in Bolivia, covering a period of 22 years (January 2000 to December 2022). The research compiled 37 peer-reviewed articles and 27 reports/documents on Bolivian regulations, focusing primarily on farmers and revealing high levels of pesticide exposure assessed through biomarkers. Overall, the manuscript presents a thorough and insightful review of the current state of pesticide exposure and its effects in Bolivia, highlighting significant gaps in regulation, enforcement, and understanding of long-term health impacts. The findings underscore the urgent need for more stringent enforcement of regulations and increased awareness and training on safe pesticide use and handling. I want to accept the article after correction of some issues mentioned below:
v The literature search, conducted in PubMed, Web of Science, Scielo, and Google Scholar, included 37 peer-reviewed articles and 27 reports/documents on Bolivian regulations. While this is commendable, the review might have benefited from a broader range of databases, including regional or national databases that might contain more localized studies not indexed in the selected databases.
v The paper highlights that a significant portion of pesticides in Bolivia are smuggled and that official data might represent just a fragment of the true status of pesticides in the country. However, the paper does not explore in detail the implications of this illegal trade, such as the types of pesticides involved, their sources, and the challenges in controlling this issue.
v The paper notes a scarcity of environmental monitoring studies in Bolivia, which is a significant gap in understanding the full extent of pesticide contamination in various environmental matrices.
v The study mentions limited research on human exposure to pesticides in Bolivia, focusing mainly on certain population segments like agricultural workers and the general population. Expanding the scope to include other potentially affected groups, such as urban dwellers or children, could provide a more comprehensive understanding of the impact of pesticides
Author Response
We are very grateful to you for the time invested in reviewing our manuscript and for the helpful comments and suggestions that helped us to clarify and generally improve our paper by pointing out missing information.
We have carefully considered each of your comments and modified the text to address the concerns raised. The guidance has undoubtedly enriched the overall content of the paper.
Please find our point-by-point responses below.
Reviewer: 2
The paper "Human Pesticide Exposure in Bolivia: A Scoping Review of Current Knowledge, Future Challenges, and Research Needs" provides a comprehensive overview of pesticide exposure in Bolivia and its various health impacts. The paper provides a comprehensive review of pesticide exposure in Bolivia, covering a period of 22 years (January 2000 to December 2022). The research compiled 37 peer-reviewed articles and 27 reports/documents on Bolivian regulations, focusing primarily on farmers and revealing high levels of pesticide exposure assessed through biomarkers. Overall, the manuscript presents a thorough and insightful review of the current state of pesticide exposure and its effects in Bolivia, highlighting significant gaps in regulation, enforcement, and understanding of long-term health impacts. The findings underscore the urgent need for more stringent enforcement of regulations and increased awareness and training on safe pesticide use and handling. I want to accept the article after correction of some issues mentioned below:
v The literature search, conducted in PubMed, Web of Science, Scielo, and Google Scholar, included 37 peer-reviewed articles and 27 reports/documents on Bolivian regulations. While this is commendable, the review might have benefited from a broader range of databases, including regional or national databases that might contain more localized studies not indexed in the selected databases.
Response 1:
Thank you for the observation, we have now included documents from different local database (press release articles and theses from University repositories) and updated number of the documents (included in the abstract and in the methodology sections) To our knowledge there are no other local databases.
v The paper highlights that a significant portion of pesticides in Bolivia are smuggled and that official data might represent just a fragment of the true status of pesticides in the country. However, the paper does not explore in detail the implications of this illegal trade, such as the types of pesticides involved, their sources, and the challenges in controlling this issue.
Response 2:
This is not the focus of the review, and due to the characteristics of informality and illegality in commerce, data on smuggling and falsification of agrochemical products are not accounted for. Even though neighboring countries have estimated these data, this has not been done for Bolivia. Moreover, there is always a concern about possible underestimation of the percentage of illegal trade. This is the only source of information that we found in relation to this topic: https://ibce.org.bo/images/publicaciones/ce-Comercio-Ilegal-de-Plaguicidas.pdf. This reference is included in the review.
v The paper notes a scarcity of environmental monitoring studies in Bolivia, which is a significant gap in understanding the full extent of pesticide contamination in various environmental matrices.
Response 3: We agree with the comment, this was one of our motivations to write the review.
v The study mentions limited research on human exposure to pesticides in Bolivia, focusing mainly on certain population segments like agricultural workers and the general population. Expanding the scope to include other potentially affected groups, such as urban dwellers or children, could provide a more comprehensive understanding of the impact of pesticides
Response 4:
We agree with your comment, we found very few articles related with children and those that we found are included. We have emphasized this challenge by adding: “These studies should be repeated and expanded to assess the current exposure levels in the general population, and especially for susceptible groups such as pregnant mothers and children” (Page 14). We have also now expanded on this aspect in the future challenges section (Page 24).
Reviewer 3 Report
Comments and Suggestions for Authors
First of all I would like to congratulate the authors on a well written review.
I only have minor changes to suggest to the manuscript.
In line 53, you need to include a reference for your statement about Chinese imports.
Change “…since many years” to "...for many years" in line 269.
My main comment concerns the concluding paragraph. Although I personally agree that prioritizing organic agriculture is extremely important, it feels that this point is thrown in at the end and comes across as naiive, ultimately doing the authors’ expertise a disservice. I would recommend that you expand this final paragraph to develop your recommendations. This could include multisectoral approaches such as research working more closely with farmers and policy makers to develop recommendations and interventions.
Author Response
We are very grateful to you for the time invested in reviewing our manuscript and for the helpful comments and suggestions that helped us to clarify and generally improve our paper by pointing out missing information.
We have carefully considered each of your comments and modified the text to address the concerns raised. The guidance has undoubtedly enriched the overall content of the paper.
Please find our point-by-point responses below.
Reviewer: 3
First of all, I would like to congratulate the authors on a well written review.
I only have minor changes to suggest to the manuscript.
- In line 53, you need to include a reference for your statement about Chinese imports.
Response 1 : As you suggest we included the reference (5) in line 63. Instituto Boliviano de Comercio Exterior. Comercio ilegal de plaguicidas en
Bolivia, un atentado a la salud, al medio ambiente y a la economía 2019 [Internet]. Vol. 28 IBCE. 2019. Available from: https://ibce.org.bo/publicaciones-descarga.php?id=2460&opcion=1
- Change “…since many years” to "...for many years" in line 269.
Response 2: As you suggest we changed “…since many years to “ …..for many years in the manuscript line 304.
- My main comment concerns the concluding paragraph. Although I personally agree that prioritizing organic agriculture is extremely important, it feels that this point is thrown in at the end and comes across as naiive, ultimately doing the authors’ expertise a disservice. I would recommend that you expand this final paragraph to develop your recommendations. This could include multisectoral approaches such as research working more closely with farmers and policy makers to develop recommendations and interventions.
Response 3: Thank you for this comment. We have followed the reviewer’s suggestion and extended the final paragraph of the conclusions (p 24):
“Finally, we conclude that prioritizing organic agriculture in training, research, and agricultural guidance is of utmost significance. Multisectoral approaches, such as closer collaboration between researchers, farmers, and policymakers to develop recommendations and interventions, can be effective in promoting food security, public health, and environmental conservation. The shift towards organic agriculture can also contribute to these goals by reducing reliance on harmful chemical inputs, not only in Bolivian agriculture but also worldwide.”